# Signatures of paracrystallinity in amorphous silicon from machine-learning-driven molecular dynamics

Louise A. M. Rosset [1], David A. Drabold [2] & Volker L. Deringer [1]✉

The structure of amorphous silicon has been studied for decades. The two main theories are based on a continuous random network and on a 'paracrystalline' model, respectively—the latter defined as showing localized structural order resembling the crystalline state whilst retaining an overall amorphous network. However, the extent of this local order has been unclear, and experimental data have led to conflicting interpretations. Here we show that signatures of paracrystallinity in an otherwise disordered network are indeed compatible with experimental observations for amorphous silicon. We use quantum-mechanically accurate, machine-learning-driven simulations to systematically sample the configurational space of quenched silicon, thereby allowing us to elucidate the boundary between amorphization and crystallization. We analyze our dataset using structural and local-energy descriptors to show that paracrystalline models are consistent with experiments in both regards. Our work provides a unified explanation for seemingly conflicting theories in one of the most widely studied amorphous networks.

Amorphous silicon (a-Si) is one of the most widely studied disordered network solids[1–4], owing in equal parts to fundamental interest and to its range of applications. In particular, a-Si has a larger band gap than its crystalline counterpart, which is useful for solar-cell heterojunctions and thin-film transistors[5,6], while its low mechanical loss makes it a candidate next-generation interferometer mirror coating material in the detection of gravitational waves using the LIGO or VIRGO instruments[7,8].

A great challenge to understanding the 'true' local structure of a-Si is that there are various preparation methods, including self-ion implantation[9], laser glazing[10], or evaporation[11], and that the structure of the resulting films depends strongly on the way by which they were made. In particular, the density[9,12], coordination environments[13,14], and the presence of voids[15,16] vary from one sample to the next. While some authors regard self-ion-implanted a-Si as the highest quality a-Si, this must be understood to be only one example of the material, albeit superbly characterized.

From foundational work in the 1930s[17,18] has emerged the currently most widely accepted model for the structure of a-Si, known as the continuous random network (CRN). The CRN model is characterized by minimal deviation from 4-fold coordination and complete absence of long-range structural order. Computations using bond-switching methods[19,20] have helped to popularize the CRN model. While a-Si cannot be experimentally quenched from the melt in bulk form[21], machine-learning- (ML-) based interatomic potentials[22] have recently enabled molecular dynamics (MD) simulations of quenching bulk a-Si at rates of $10^{11}$ K s$^{-1}$ (ref. 23) and slower[24]. Such rates are comparable to those used in laser quenching experiments[25].

Despite the simplicity of the CRN model, and the fact that it is now widely seen as the preferred way to describe a-Si[1], this model is not without challenges. The main argument against the CRN model is that it fails to capture the degree of medium-range order seen in fluctuation electron microscopy (FEM) experiments on a-Si[26]. Instead, an alternative explanation consistent with FEM data has been proposed[26,27], known as the 'paracrystalline' model. The latter is defined as a strained nanocrystal embedded in an amorphous CRN matrix, without sharp grain boundaries[26]. Such paracrystalline structures have recently been synthesized and experimentally and computationally characterized for

[1]Department of Chemistry, University of Oxford, Oxford, UK. [2]Department of Physics and Astronomy, Ohio University, Athens, OH, USA.
✉e-mail: volker.deringer@chem.ox.ac.uk

the lighter homolog, elemental carbon[28]. However, the paracrystalline model for a-Si conflicts with other experimental data[2,3] and for many only qualifies as a mixed-phase material[1,29]. For some authors, the answer lies in an intermediate network between disordered and ordered Si[29] which would explain findings related to the low-energy excitations of a-Si[30], while others argue from calorimetric data that there exists a configurational gap between amorphous and crystalline networks[31,32]. In short, the long-standing CRN vs paracrystalline debate has not been fully resolved[33–35].

In the present study, we probe the limit between amorphization and crystallization of simulated melt-quenched Si. We systematically sample the configurational space of a-Si with an accurate and efficient teacher–student ML approach[36] (Methods), which allows us to explore the existence of a middle ground between fully disordered and crystalline structures. Both system size and simulation time, unlocked by efficient ML methods[23,36], are key to a full exploration of competing phases and microstructures. The results lead us to propose a revised paracrystalline Si model that is consistent with high-quality structural and calorimetric experimental data. We characterize these paracrystalline clusters, and quantify structural and energetic properties of a-Si models over the range from disorder to order, thereby allowing us to gain unprecedented insight into the coexistence of the CRN and paracrystalline phases. In so doing, we show that realistic and experimentally compatible models of a-Si are able to accommodate a small but significant degree of local paracrystalline order, whilst overall remaining a disordered network.

## Results

### A continuous range from disorder to order

We created a library of a-Si structural models in MD simulations with a systematically varied range of parameters. Specifically, we performed melt-quench simulations for four system sizes (64, 216, 512, and 1,000 atoms) with a uniform range of densities between 2.1 and 2.5 g cm$^{-3}$, over four quench rates of $10^{13}, 10^{12}, 10^{11},$ and $10^{10}$ K s$^{-1}$. To obtain a set of uncorrelated structures, we only take the final frame from each melt-quench simulation. This results in a dataset of 3069 unique structures ($\approx$ 1.3 million atoms). We note that in this part of the study, we focus on relatively small simulation cells on purpose; we will subsequently describe larger (100,000 atoms per cell) structural models.

Our dataset (Fig. 1) contains structures ranging from highly disordered to very close to the crystalline form (c-Si). We characterize the dataset by plotting the computed excess energy, $\Delta E$ (relative to c-Si), against a measure for the similarity to the crystalline reference, where 1 is identical (see Section "Methods"). We define structures as being either fully CRN-like, or paracrystalline, or polycrystalline using polyhedral template matching (PTM)[37]. Some 64-atom structures fully crystallized and formed strained diamond, shown in gray in Fig. 1b.

The fact that our dataset ranges almost smoothly from disorder to order (left → right), both energetically and topologically, challenges the hypothesis of a configurational energy gap between c-Si and a-Si[32]. The paracrystalline structures populate the energetic middle ground between the CRN-like and polycrystalline configurations—which also challenges the initial theory of a higher-energy paracrystalline phase that could be annealed to yield a CRN[26]. While our dataset is relatively uniformly distributed, we observe a lower density of structures at the

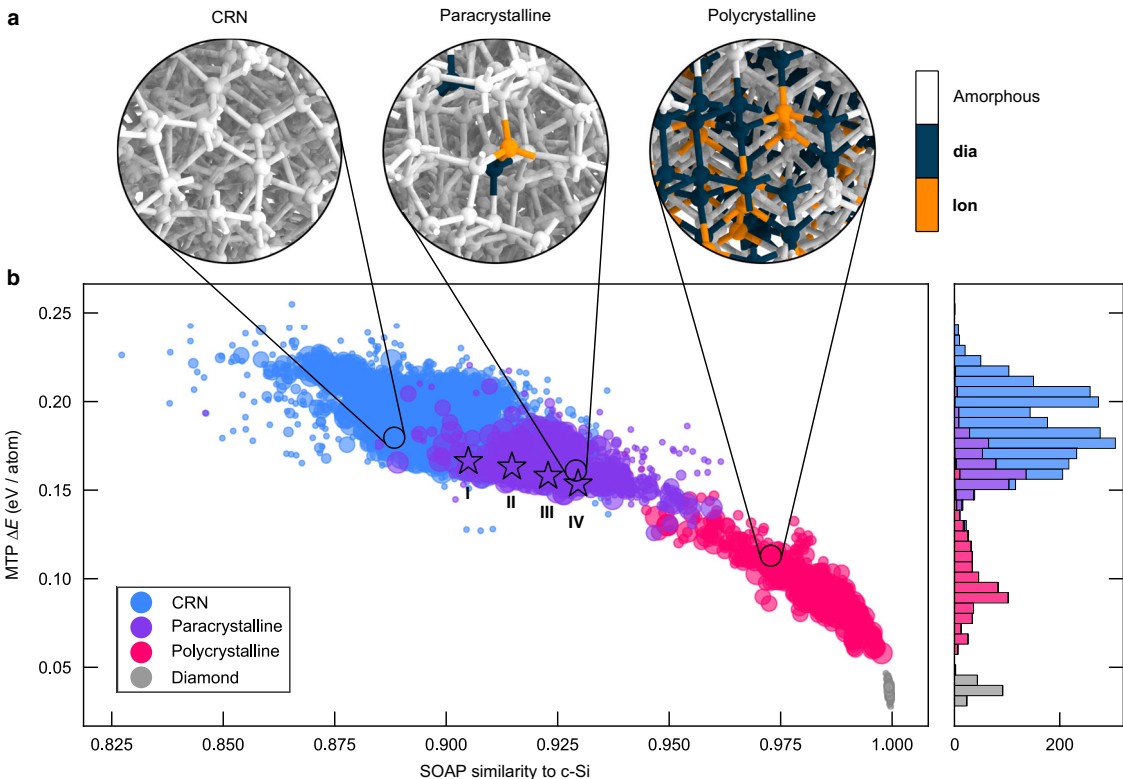

**Fig. 1 | A comprehensive dataset of disordered Si structures. a** Ball-and-stick rendering of representative structures from three categories, viz. continuous random network (CRN, left), paracrystalline (center), and polycrystalline (right). Polyhedral template matching was used to characterize atomic environments: blue indicates cubic-diamond-like environments (**dia**), orange indicates hexagonal-diamond-like (**Ion**) ones, and white indicates atoms that do not fall within one of the defined categories (see Methods for details). **b** A map of similarity to diamond-type Si against the predicted excess free energy (see Section "Methods"). The marker sizes are proportional to the number of atoms in the respective structure. Stars indicate a selection of four structures of increasing paracrystallinity, labeled **I** to **IV**, further discussed in Fig. 2. A stacked histogram of the energies is shown on the right, using the same vertical axis. The distributions in (**b**) indicate that the dataset spans structures from CRN- to diamond-like, encompassing a smooth range from disorder to gradual order. Source data are provided in the Source Data file.

paracrystalline–polycrystalline transition, around 0.14 eV on the energy histogram in Fig. 1. This corresponds to a deficit of structures with locally 'crystal-like' environments between 15 and 40%. These structures are likely absent from our dataset due to fast crystal-growth kinetics post nucleation, resulting in fewer structures with small crystalline grains. We note that the 64-atom a-Si structures (small markers) scatter widely in the plot of Fig. 1, but sample a rather similar configurational space to the other system sizes (Supplementary Fig. 10).

### Defining and characterizing paracrystalline structures

While the paracrystalline category is intermediate between the CRN and polycrystalline ones, it shares significant topological and energetic overlap with the former. We select four paracrystalline structures of 1000 atoms in the overlapping range, denoted **I** to **IV**, for more detailed analysis. These structures are increasingly paracrystalline, as reflected by their percentage of diamond-like environments of 0.2% (**I**), 0.8% (**II**), 2.4% (**III**) and 4.5% (**IV**); they are visualized in Supplementary Fig. 13. In Fig. 2, we use established indicators of short- and medium-range order to study these four structures. The radial distribution functions (RDFs) (Fig. 2a) are overall similar, with a well-defined valley between the first and second peak, indicating well-relaxed structures. The most relevant aspect in the context of paracrystallinity is the region between the second and third peaks, where experiments[14,38–40] showed a small but notable enhancement at about 4.5 Å. Our series of models shows the gradual emergence of such a feature; the ratio between local maximum (at $\approx 4.5$ Å) and local minimum (at $\approx 5.0$ Å) is 1.08 for **I** but 1.47 for **IV**. Hence it is absent from the structure closest to CRN but replicated in the more paracrystalline structures. This feature has been attributed to a preferential orientation in the dihedral bond-angle distribution[14,39], for which we show computed results in Fig. 2b. As paracrystallinity increases, the distribution sharpens while staying smooth—disagreeing with the claim that the RDF feature is only affected by the smoothness of the dihedral-angle distribution and not by its sharpness[41]. Our results are qualitatively consistent with previous reports of paracrystalline signatures in the dihedral-angle distribution[42,43]. The shortest-path ring distribution (Fig. 2c) also

mirrors the increasing degree of ordering from **I** to **IV**: 6-membered rings, characteristic of c-Si, become more abundant with paracrystallinity.

To further understand the diamond-like environments and their characteristics, we analyze clustering trends in our dataset. We sort all structures of 1000 atoms with diamond-like environments, that is, all structures but the CRN category, into bins according to their SOAP similarity to c-Si, rounded to the second decimal place. We present statistics of the average number of clusters and the average cluster size, as a function of the SOAP similarity to c-Si, in Fig. 3.

Figure 3a shows that increasing SOAP similarity to c-Si between 0.89 and 0.92 does not lead to a sharp spike in the cluster size, but rather a very slow increase from an average size of 1.0 to 3.73 atoms per cluster. At higher SOAP similarity values (higher crystallinity), the count of atoms per cluster rapidly increases and exceeds the experimental estimate of the critical nucleus size of 40–60 atoms[44]. Increasing the SOAP similarity to c-Si rapidly increases the number of clusters per structure, which reaches a maximum at a SOAP similarity of 0.95, as shown in Fig. 3b. Beyond this, clusters interconnect and the number of clusters per structure drops. At a SOAP similarity of 0.99, one cluster spans almost the entirety of the structure.

At low SOAP similarities, clusters exist at small distances from one another (Fig. 3c). As more clusters appears and grow, neighboring diamond-like environments join. There are fewer diamond-like atoms within a short neighborhood as neighbors have already joined the cluster—the distance between clusters increases. We further hypothesize that as clusters grow larger, the surrounding CRN matrix becomes increasingly strained, and clusters effectively repel each other to larger inter-cluster distances. At very high SOAP similarities, single clusters have almost grown to the entire structure, and we observe a return of very short inter-cluster distances.

### Energetic fingerprints

Our analysis so far has established that the paracrystalline structures are structurally reasonable. The next step is to compare them directly with existing CRN models and to differentiate them from polycrystalline Si. In addition to structural information, it is important to consider energetic arguments. In Fig. 4, we therefore focus on the local-energy fingerprints which can be derived from machine-learned atomic energies (see Section "Methods"). It was shown previously that such an approach can help to map out the space of disorder and local order in monolayer amorphous carbon[45], for which the distinction between CRN and (para-) crystallite descriptions has also been explored[45,46]. The present analysis in Fig. 4 hence takes us conceptually from a canonical disordered 2D system, amorphous graphene, to the canonical 3D case, which is a-Si.

For each of the three representative structures shown in Fig. 1a, we represent the individual atomic environments therein as circles in Fig. 4. We plot their computed excess energy, $\Delta E$ (relative to c-Si), averaged over their nearest neighbors, against a structural metric that quantifies how similar a given atom is to cubic-diamond-like Si (SOAP; Methods). We color-code the points based on Common Neighbor Analysis (CNA; Methods).

Figure 4 allows us to characterize the three fundamental forms that disordered silicon can take. The CRN structure shows only amorphous-like atomic environments, as expected. The energy histogram (horizontal axis) and SOAP similarity histogram (vertical axis) both show a single peak with a long tail. For the paracrystalline structure, some **dia** and **lon** environments are identified by CNA, but the majority of atomic environments are still amorphous-like. These diamond-like environments are far from the ideal diamond environment (star); they are not clustered together but distributed among the amorphous environments. The tails in both histograms are shorter, indicating that the amorphous environments in the paracrystalline structure do not suffer from additional strain from the presence of the

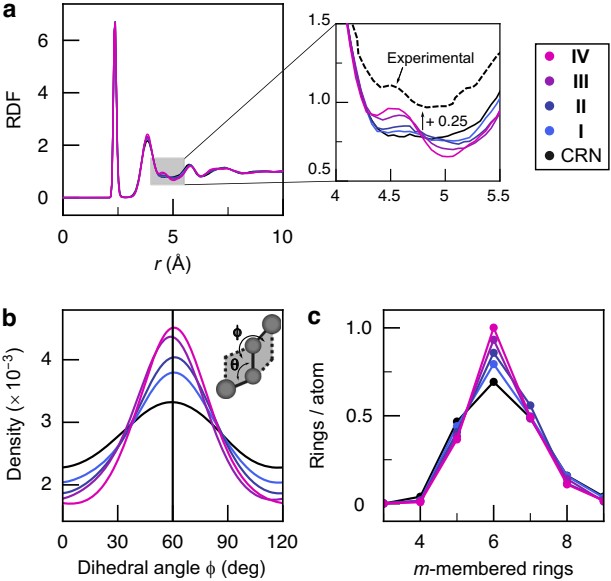

**Fig. 2 | Characteristics of medium-range order.** We study four structural models of increasing paracrystallinity (**I–IV**) as well as the CRN structure shown in Fig. 1a. **a** Radial distribution function, and corresponding inset with experimental RDF from ref. 38. **b** Dihedral-angle distribution with a schematic indicating the definition of $\phi$. **c** Distribution of $m$-membered shortest-path rings. Source data are provided in the Source Data file.

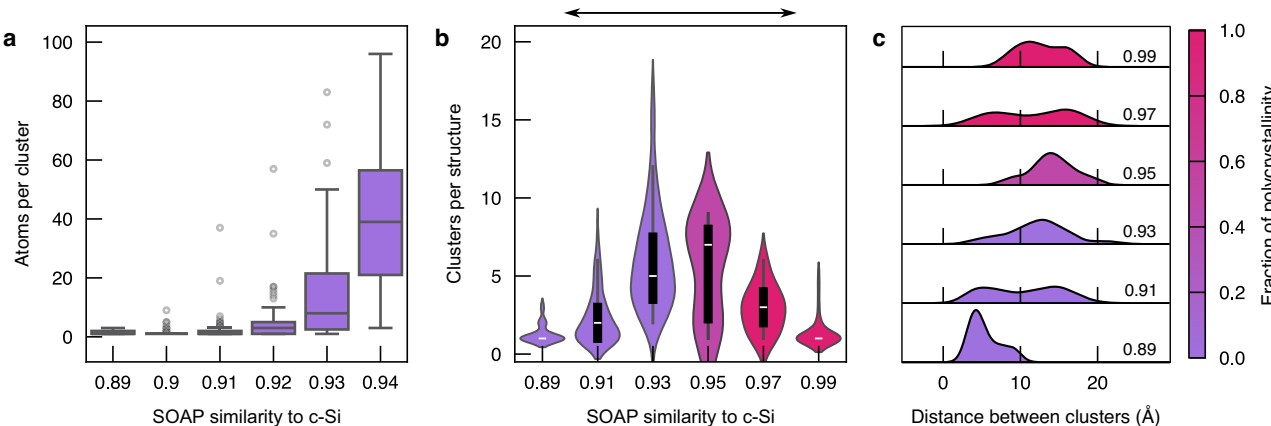

**Fig. 3 | Clustering of diamond-like environments in structures of 1,000 atoms.** **a** Count of the number of atoms per cluster, presented for structures with different SOAP similarity values from 0.89 to 0.94, a range which contains only para-crystalline structures. The boxes show the interquartile range (IQR), with the median shown as a line in the box, and the whiskers show the range of the dis-tribution within 1.5 × IQR and exclude outliers, or points lying beyond this threshold, shown in gray. **b** Count of the number of clusters per structure, pre-sented for every other SOAP similarity values ranging from close to CRN (0.89) to diamond (0.99). The violin plots present a box plot of the distribution inside a kernel density estimate of the distributions, with kernel bandwidth of 1. **c** Distribution of shortest distance between clusters. For all three plots, structures were sorted into categories (binned) according to SOAP similarity to c-Si, and the colormap indicates the total fraction of polycrystalline structures within a given bin. The threshold between paracrystalline and polycrystalline structures is 15% of locally crystal-like atoms, as determined by Polyhedral Template Matching. Source data are provided in the Source Data file.

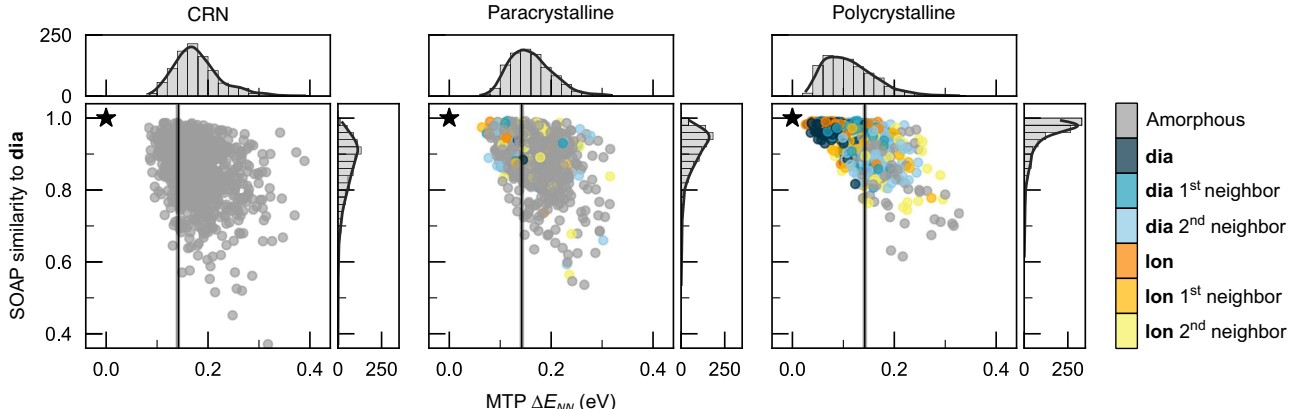

**Fig. 4 | Energetics of disordered Si structures.** Scatter plot of the ML-predicted atomic energy relative to cubic-diamond-type Si (**dia**) averaged over nearest neighbors against the atomistic SOAP similarity to **dia**, colored by adaptive Com-mon Neighbor Analysis. A star indicates the ideal **dia** environment. Histograms of the total distribution and kernel density estimates are shown for each axis. Vertical lines indicate the experimentally measured heat of crystallization, with gray shad-ing corresponding to the standard deviation[11]. Source data are provided in the Source Data file.

localized diamond-like environments. Finally, for the polycrystalline structure, diamond-like environments are distinct from amorphous ones in both energy and structure. **dia** and **Ion** environments are much closer to the ideal diamond environment than those in the para-crystalline structure are. The energy and SOAP histograms are char-acterized by two contributions, one from diamond-like and one from amorphous-like environments. Thus, the paracrystalline Si structures are comparable to the CRN ones, and can be delineated from the polycrystalline structures. We can ascertain that they are disordered, with localized crystal-like signatures.

The experimentally measured heat of crystallization, $\Delta H = 0.142$ eV/atom[11], is plotted alongside our ML local atomic energies in Fig. 4. The paracrystalline structure agrees very well with these calorimetric data, where the CRN model is more energetic and the polycrystalline model is too stable compared to $\Delta H$. The paracrystalline structure also provides better agreement to $\Delta H$ than previous CRN models in the literature[24].

## Device-scale models

While our dataset provides valuable insight into the middle ground between fully disordered and crystalline silicon, the fact that we have used relatively small system sizes limits the comparability to experi-mental data. We therefore turn to a study on a larger length scale, viz. > 10 nm, which is relevant to a-Si-based devices such as photodiodes or light sensors[47–49], as well as chalcogenide-based memory devices[50]. We prepare para- and polycrystalline models with cell lengths of about 12 nm using the same protocol as for the dataset, yielding models with 0.8% and 62.3% of diamond-like environments, respectively. We com-pare against the structural model of ref. 4 which had been created in simulations of the same type but driven by the teacher model, Si-GAP-18, and has 0.3% of diamond-like environments. These structures are shown side-by-side in Fig. 5. The structure factor, $S(q)$, for each model is plotted together with high-quality experimental data from ref. 38. The latter are well reproduced by the model with the lowest paracrystallinity[4]—but also by a more paracrystalline model, which is

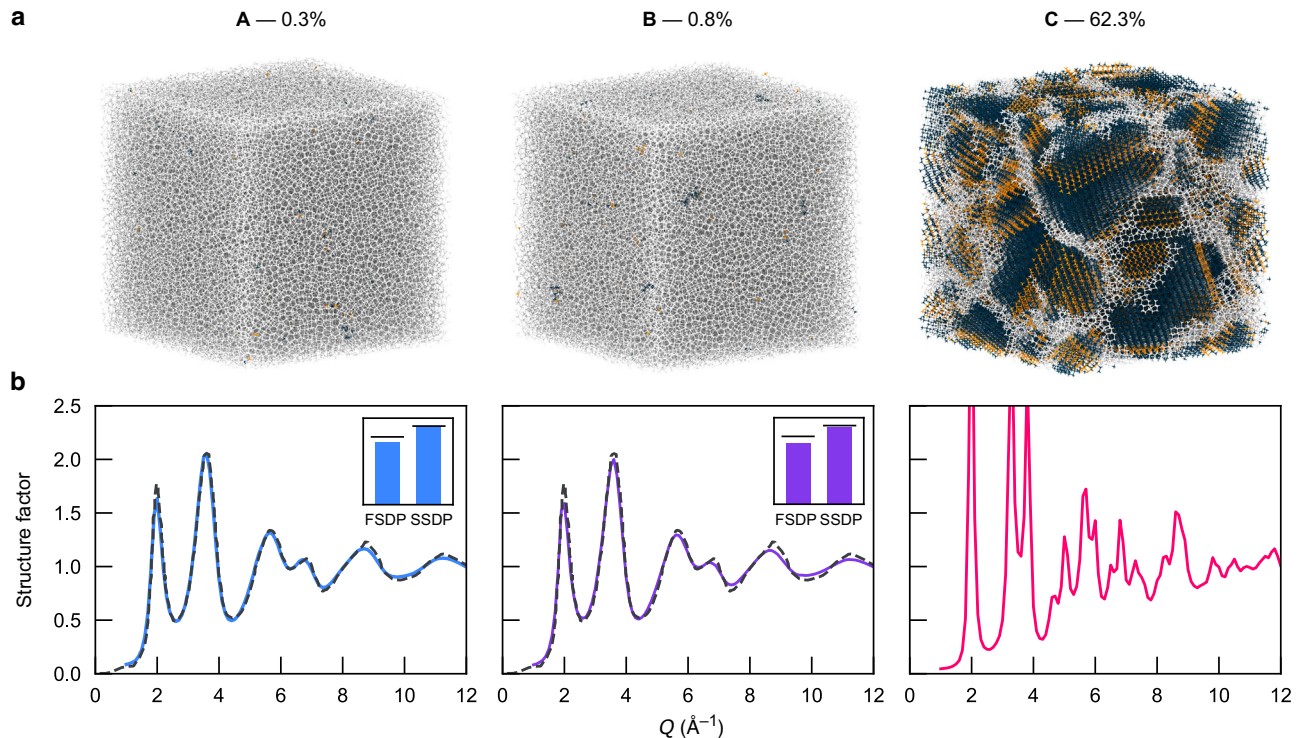

**Fig. 5 | Three a-Si structural models of 100,000 atoms with increasing degree of crystalline order.** The first model, labeled A, is taken from ref. 4, while the other two were generated as part of the present study by melt-quenching at $10^{11}$ K s$^{-1}$ (B) and $10^{10}$ K s$^{-1}$ (C), respectively. **a** Structure visualizations color-coded by PTM as in Fig. 1. **b** Normalized structure factor for each structure, computed using the DEBYECALCULATOR package[70]. Black dashed lines indicate the experimental data from ref. 38. Insets show the agreement of the predicted first and second sharp diffraction peaks (bars) with experimental data from ref. 38 (black lines). Source data are provided in the Source Data file.

just as compatible with the experimental data. This implies that localized order can retain model agreement with experimental data, but only a small degree of crystallinity is beneficial as the polycrystalline model shows large crystalline regions leading to unwanted peaks in the structure factor.

## Discussion

We have systematically sampled the configurational space of Si, from fully disordered CRN-like networks to the diamond-type crystal, with extensive ML-driven atomistic simulations. Our results point toward a revised model for paracrystalline Si, at the limit between amorphization and crystallization, characterized by localized diamond-like neighborhoods that affect medium-range order. Paracrystalline structures show better agreement with high-quality experimental data for medium-range structural order and energetics than do previously proposed models. We note that while high-quality experiments are typically carried out on ion-implanted a-Si samples, laser-glazed a-Si is much closer to the melt-quenched samples generated by MD simulations. Further experimental work on laser-glazed a-Si could provide a closer basis for comparison, informing future theoretical and computational studies.

Our work opens important new avenues of exploration. As our dataset spans an essentially complete range of disorder, it is of interest to explore emergent phenomena unique to disordered matter such as the process of photodegradation known as the Staebler–Wronski effect[51,52]. Much research has been conducted on a-Si and a-Si:H, driven partly by photovoltaic applications[47]. Following much interest in hydrogenated a-Si cells around 2000, there is now renewed focus on inexpensive a-Si:H cells for low-power applications, and as a component of tandem solar cells[53]. So-called 'protocrystalline' photovoltaics[54] are presumably hydrogenated variants of the paracrystalline phases that we study in the present paper, and they exhibit improved photo-

stability relative to amorphous materials. Evidently, progress on the atomistic origins[55] of the Staebler–Wronski effect requires large and realistic structural models. Indeed, a natural starting point could be to hydrogenate suitable models reported in the present paper, e.g., using a recently developed GAP ML potential for a-Si:H[56].

Two-level tunneling systems (TLSs), described as the tunneling between neighboring minima in the potential-energy landscape of amorphous materials, are also of fundamental interest for a-Si because they offer an explanation for low-energy excitations found at low temperatures[34,57]. A proposed origin for TLSs is nanoscale heterogeneity in the microstructure, taking the form of local order[30]—such heterogeneity has been out of range for direct quantum-mechanical simulations, but is accessible using ML[58]. Systematically searching for perturbations that result in pairs of nearly identical amorphous configurations along the dataset's range from disorder to order could help to determine what extent of structural disorder in the network is required to observe tunneling[59]. Hence, our work provides a high-quality dataset for further exploration of outstanding research questions related to a-Si, and more widely it exemplifies the role of ML in understanding fundamental phenomena in disordered materials.

## Methods

### Teacher–student potentials

The simulations in this work are based on a teacher–student machine-learning approach[36]: distilling an accurate, but comparably slow 'teacher' ML potential (Si-GAP-18; ref. 22) into a faster 'student' model, here using the Moment Tensor Potential (MTP) approach[60]. We use the $M''_{16}$ model of ref. 36, which provides accuracy approaching that of Si-GAP-18 within the target domain (a-Si), whilst being > 100 times faster. The teacher model has been extensively validated against experimental data for ambient and high-pressure a-Si[4,23], and the student model has enabled recent studies of coordination defects[61]. We

provide further comparison with the Si-GAP-18[22] and Si-ACE-21[62] ML potentials in Supplementary Information, Supplementary Figs. 4–6.

### Structural analysis

We classify structures as being either fully CRN-like, or paracrystalline, or polycrystalline using polyhedral template matching of atomic environments (PTM; RMSD cutoff of 0.1; ref. 37) as implemented in OVITO[63], with the following criteria: (i) if a structure contains no locally 'crystal-like' atom, it is classified as fully CRN-like (blue in Fig. 1); if it contains (ii) fewer or (iii) more than 15% of locally 'crystal-like' atoms, it is classified conversely as paracrystalline (purple) or polycrystalline (magenta). The 'polycrystalline' category is diverse, from large crystalline grains in an amorphous matrix to diamond structures with stacking faults. We justify our choice of threshold as part of Supplementary Note 1.

For the analysis of local atomic environments, we employ two complementary techniques. First, we use the Smooth Overlap of Atomic Positions (SOAP) kernel[64] to quantify the similarity to the ideal diamond-type structure on a scale from 0 (dissimilar) to 1 (identical to within the cutoff radius), as done in previous work on a-Si, setting $\zeta = 4$[24,36]. Second, we use Common Neighbor Analysis (CNA)[65] to identify the similarity to prototype structure types (specifically, **dia** and **lon**), as detailed in ref. 66, and used in ref. 28. It is implemented in OVITO,[63] We report a comparison between descriptors as part of Supplementary Note 1.

### Energetic analysis

In many ML-based interatomic potentials, including the MTP framework, the total energy of a cell is constructed as the sum of the ML-learned individual atomic energies[67,68], viz. $E = \sum_i E_i$. The distribution of such atomic energies has been shown to reveal the local stability of atoms in systems ranging from a-Si[24] to superionic conductors[69]. We further take the local atomic energies averaged over nearest neighbors, similar to a study of amorphous graphene[45], here within a cutoff of 2.85 Å[24].

## Data availability

Data supporting this study are openly available at https://github.com/lamr18/aSi-data and at https://doi.org/10.5281/zenodo.14203730. Source data are provided with this paper.

## Code availability

Software for ML-driven MD simulations was used as provided by the authors and described in the Methods section. The $M_{16}''$ potential is openly available at https://zenodo.org/records/7003068, and was used alongside the MLIP package (https://mlip.skoltech.ru/download) to run Molecular Dynamics simulations in LAMMPS. Custom-written code to carry out analyses is available at https://github.com/lamr18/aSi-data; a version has also been deposited together with the research data at https://doi.org/10.5281/zenodo.14203730.

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

## Acknowledgements

We thank S. R. Elliott, J. D. Morrow, and M. Wilson for helpful comments on the manuscript. This work was supported through a UK Research and Innovation Frontier Research grant [grant number EP/X016188/1] (V.L.D.). We are grateful for computational support from the UK national high performance computing service, ARCHER2, for which access was obtained via the UKCP consortium and funded by EPSRC grant ref EP/X035891/1 (https://www.ukcp.ac.uk/pmwiki.php/UKCP/Acknowledgement) (V.L.D.).

## Author contributions

L.A.M.R. and V.L.D. designed the study. L.A.M.R. carried out all computational work with guidance from V.L.D. All authors (L.A.M.R., D.A.D., and V.L.D.) contributed substantially to discussions and to the interpretation of the data. L.A.M.R. drafted the manuscript, and all authors contributed to its writing and approved the final version.

## Competing interests

The authors declare no competing interests.
