## [Transparent Peer Review file · Nature Communications]

Signatures of paracrystallinity in amorphous silicon from machine-learning-driven molecular dynamics

Corresponding Author: Professor Volker Deringer

Version 0:

Reviewer comments:

Reviewer #1

(Remarks to the Author)

This manuscript describes physically realistic modeling of amorphous silicon (a-Si) and paracrystallinity which may be present in the network as well as crystallization of the network. This type of modeling, validated by experimental observations, provides the framework for further investigations in the modifications of a-Si networks, such as through incorporation of hydrogen, dopant atoms, or group IV elements (C, Ge, Sn). The machine learning approach expands the capabilities of modeling to more closely describe larger scale systems, such as these amorphous networks. I recommend the manuscript be accepted for publication after the following points are addressed:

1) In the conclusion, application of this approach to other phenomena such as Staebler Wronski effect are mentioned. Hydrogenated amorphous silicon has been substantially studied, leading to other nomenclature such as “protocrystalline” and “edge” material. Descriptions of how these materials (and others) fit within the approach here should be added.

2) Diamond like atomic environments are identified in the model with variations in percentages tracked with short and medium range order. With increased diamond like percentages, does the interconnectivity of the diamond like regions increase or do they remain relatively isolated? If interconnected, over what length scale?

Journal Response Questions:

[What are the noteworthy results?] The noteworthy results consist of the development of a model spanning continuous random networks for amorphous silicon, paracrystallinity in silicon, and through to polycrystalline silicon. This includes energetic concerns relative to similarity of different networks to crystalline silicon, predicted medium range order compared to experiment, and structure factors. 100,000 atom models are developed so that the behavior of amorphous networks closer to that of reality (> 10 nm x 10 nm x 10 nm) length scales can be evaluated.

[Will the work be of significance to the field and related fields? How does it compare to the established literature? If the work is not original, please provide relevant references.] This work is relevant to the field of amorphous silicon and amorphous materials in general. It provides physically realistic models that bridge the gaps from truly random networks through those that have some degree of crystallinity and shows that there is a continuum of behavior. It builds from established literature and previous work by these authors.

[Does the work support the conclusions and claims, or is additional evidence needed?] The results support the conclusions drawn. Theoretical models are validated by some experimental evidence.

-[Are there any flaws in the data analysis, interpretation and conclusions? - Do these prohibit publication or require revision?] The previous comments involve more description of the literature in the amorphous silicon system and description of the isolation or interconnectivity of the diamond like regions remain to be addressed prior to publication.

[Is the methodology sound? Does the work meet the expected standards in your field?] The methodology builds from that established in the modeling of amorphous networks and expands it to larger size models. This is crucial to more realistically identifying the characteristics of disordered materials over the length scales where the materials are fabricated and utilized in applications.

[Is there enough detail provided in the methods for the work to be reproduced?] Yes, there is sufficient detail to reproduce the

methodology.

Reviewer #2

(Remarks to the Author)

This is an excellent paper of wide interest and importance as it addresses the fundamental structure of an important amorphous semiconductor. Silicon is also one of the most well-studied amorphous materials, yet ambiguity remains about the basic structure – is it a continuous random network (CRN) or does it have a higher degree of order? The authors, who are well regarded in the field of computational modeling and amorphous materials, use an innovative machine-learning approach to examine a very wide range of “quenched” structures and reveal an unexpected band of paracrystalline structures with a lower energy than the continuous random network. Their arguments are convincing that their work addresses a number of experimental results and adds substantial weight to the claim that the CRN is not the lowest energy state for amorphous silicon. I believe this work should be published in Nature Communications as it is a very fundamental and important new result.

A couple of small questions to be considered:

- a) Is figure 1 a plot of the enthalpy or the free energy?
- b) While the modeling represents quenching from the melt, the comment about the deficit of structures between 5% and 14% crystal environments suggests that growth is occurring during the process. Are the authors able to comment on the presence or absence of a “critical nucleus size” for crystalline Si embedded in amorphous silicon?
- c) I recognize that this is the first short publication of an exciting new result, but it would also be interesting to see some information on the physical size of the paracrystalline domains. This of course, could be covered in a future work - its just a helpful (I hope!) suggestion.

In this reviewer's opinion there is no need for subsequent review – could be published as is but the authors can consider my suggestions.

Reviewer #3

(Remarks to the Author)

In this manuscript, the authors generated some configurations of a-Si containing different fractions of crystal- or diamond-like local environments, simply using different cooling rates and starting from different density, based on an MTP potential already published.

Then, they classified these configurations into three categories, i.e., fully disordered (CRN), partially crystallized (the so-called paracrystalline) and crystallized (polycrystalline), based on the thresholds they chose arbitrarily (as there is no any physical reason for their choice of threshold values in the manuscript).

First of all, I did not see that the current work presents any surprising results or new physical insight. It is well-known that the structure of any amorphous solids strongly depends on processing conditions. One even can find crystal-like local environments in undercooled liquids. In addition, when metastable amorphous solids crystallize into stable crystals, the change in structure (fraction of crystal-like environments) is definitely continuous. Therefore, one certainly can generate/observe amorphous configurations with different fractions of crystal-like local environments under different processing conditions either in real experiments or in simulations based on either ML or classical potentials. So, the main discovery presented in the current work is not a surprise.

All the methodologies employed in the current work are already published in literature. Thus, this work did not present any new methodology.

So I can not support this work to publish on Nature Communications.

And below are some technical issues.

The authors argued that their dataset shown in Fig. 1 has high quality. But they just relaxed these configurations at 300 K for a very short time (10 ps) after fast quenching. These configurations are probably unstable or just some transient states. So why we need to care about the results based on this dataset?

According to Fig. 4, under the same cooling rate of $1e11$ K/s, using the two different potentials (GAP and MTP), the fraction of crystal-like atoms is different by 0.06%. One can understand this difference is due to artifact (the difference b/t the two ML potentials).

By the way, is there any difference in nucleation rate of silicon melts b/t the two ML potentials? For the GAP potential, nobody observed/reported crystallization from silicon melts. Interestingly, the authors observed crystallization from silicon melts with the MTP potential. Seems there is big difference in nucleation rate b/t the two ML potentials. If so, why should we believe the results based on the MTP potential. For silicon, an efficient ACE potential trained on the same dataset is also available, see npj Comput Mater 7, 97 (2021). Will the ACE potential give another different result?

Actually, the difference $> 0.06\%$ in the fraction of crystal-like atoms can be observed in simulations with an identical potential and the same cooling history but quenching from different liquids at the same high temperature. Therefore, it is trivial to discuss two configurations with a difference by 0.06% in the fraction of crystal-like atoms.

When judging if an atom is crystal-like or not, they used PTM. This method is not very reliable as the results strongly depends on the cutoff of RMSD. The widely used method based on physically well-defined bond orientational order parameter, e.g., as used in Nat. Commun. 15, 368 (2024), should be more reliable.

Also, the choice for the threshold of fraction of crystal-like atoms used in separating paracrystalline and polycrystalline is pretty arbitrary.

The authors claimed their simulations with 100,000 atoms is realistic. But, in length scale, I feel billion-atom MD simulations, e.g., as conducted in Nat Commun 8, 10 (2017) by Shibuta, Y et. al., is more realistic.

Version 1:

Reviewer comments:

Reviewer #1

(Remarks to the Author)

The authors have satisfactorily addressed my previous comments. This manuscript is now ready for publication in my opinion.

The added information in the manuscript in the Discussion section addresses the comment regarding protocrystallinity and the Staebler Wronski effect.

The added information in the manuscript (Figure 3) and supplemental information (Section S2.4) addresses the previous comment regarding interconnectivity of diamond like regions.

Reviewer #3

(Remarks to the Author)

Thanks the authors for clarifying the questions they are seeking to address.

From fully disordered to crystalline states, the structural variation, at least the time evolution of the fraction of local crystal-like environments, is definitely continuous, because it is impossible for an amorphous state to turn into crystalline state instantly, in configuration space. If this really needs a specific demonstration, I agree that the authors made it here. But I don't think that the results deserve a publication in Nature Communications. It might be a question long time ago, but now the answer is very obvious.

If the authors were demonstrating that there is a middle metastable state, i.e., so-called paracrystalline, between fully disordered and crystalline states, this would be a very interesting question. However, I do not think that the authors showed convincing evidence. To demonstrate it, one should show there is an additional basin in the middle of the free energy pathway from fully disordered to crystalline states, or show there is discontinuity in the evolution of one thermodynamic variable during the transitions from fully disordered to paracrystalline and from paracrystalline to crystalline.

The structural variation is continuous from fully disordered to crystalline states. I did not see any physical reason to pick a threshold value for a structural parameter to define an additional state, as the authors did in the manuscript. Following this logic, one could choose multiple thresholds to define multiple middle states in between fully disordered and crystalline states. Then more papers can be published.

If the authors are arguing that the amorphous model containing some fraction of local crystal-like environment shows better match to experimental data, I don't feel it is a big deal. Because one even can find small amount of local crystal-like environment in undercooled melts, it is natural to see some local crystal-like environments in glasses. It might be interesting to debate whether amorphous silicon is fully disordered or so-called paracrystalline many decades ago, now it is well understood that glass structure depends on processing history. The current results might be suitable to publish on other journals, but not Nature Communications.

I doubted that configurations are probably unstable or just some transient states because they just relaxed 10 ps. Then the authors show there is no big difference even after relaxing for 1 ns. I agree that 1 ns is long for MD simulations especially those based on ML potentials. But 1 ns is far shorter than physical time for real materials. It may be hard to use straightforward MD simulations to give a convincing answer.

The authors argue that BOO is not as good as PTM in identifying diamond-like environment from amorphous silicon. But the authors did not provide sufficient detail of the BOO they used. And I did see there is a big gap in the distribution of a single BOO between amorphous silicon and diamond crystals in the reference I mentioned in my first comments.

Reviewer #4

(Remarks to the Author)

Per the Editor's request, I will focus exclusively on the review comments provided by the third reviewer and will refrain from making recommendations on the manuscript's suitability for publication in Nature Communications.

Regarding the third reviewer's questions, I will specifically address the following:

Is thermodynamic quantification of the paracrystalline state necessary beyond the PTM method?

The reviewer raises an important point: demonstrating that the paracrystalline state is a thermodynamically metastable phase (e.g., through free energy analysis) would indeed constitute groundbreaking work. That said, this manuscript approaches the issue from a structural perspective. The authors present a realistic modeling framework to interpret the structure of amorphous silicon (a-Si), introducing the concept of paracrystallinity or medium-range crystalline order as an extended configuration of the amorphous phase (as opposed to the widely accepted CRN structure).

The observed paracrystallinity likely originates from a nucleation process with a vanishingly small nucleation barrier in the supercooled liquid state. Due to kinetic constraints, subcritical nuclei (comprising tens of atoms) are frozen in local energy minima, persisting as a metastable "paracrystalline" state at 300 K. This state is unstable at elevated temperatures (e.g., near the glass transition temperature). It is unlikely that a thermodynamic variable exists to distinguish the paracrystalline state as an intermediate phase with a discrete energy minimum separating the crystalline and continuous random network (CRN)-type amorphous phases.

From a thermodynamic perspective, the paracrystalline state extends the configurational space of CRN-type a-Si. Transitions to polycrystalline Si require thermal activation. This paracrystalline state may persist for extended timescales at 300 K (e.g., 10 ps, 10 ns, or longer), but it remains metastable under these conditions and transitions to instability at higher temperatures (i.e., slower cooling rates).

In summary:

The paracrystalline structure of a-Si results from a kinetically constrained nucleation process. It is metastable at 300 K but becomes unstable at elevated temperatures. The existence of a thermodynamic variable that could definitively classify the paracrystalline state as an intermediate phase is highly unlikely. Within this context, the authors' efforts to interpret the structure seem sufficient.

Lastly, I observed that the MTP potential shows noticeable energy discrepancies compared to the other potentials used (Figure S5).

Response to Reviewers' Comments for "Signatures of paracrystallinity in amorphous silicon"

Louise A. M. Rosset¹, David A. Drabold², and Volker L. Deringer¹

¹Department of Chemistry, University of Oxford, Oxford, UK

²Department of Physics and Astronomy, Ohio University, Athens, OH, USA

We thank all three reviewers for their careful evaluation of our manuscript. We were happy to read the very positive feedback by Reviewers #1 and #2. We also appreciate the critical comments by Reviewer #3, who has identified points that we need to justify more clearly—and so they have helped us to improve the manuscript substantially.

In brief, the main changes are:

- We further characterized the diamond-like local environments which are central to our work—in particular, looking at clustering trends across the dataset, in line with questions by Reviewers #1 and #2. We added a figure to the main manuscript (Fig. 3) and further analysis in Tables S2 and S3.
- We strengthened the technical aspects of the work by “cross-checking” with an ACE ML potential,^{R1} which is trained on the Si-GAP-18 dataset (see pp. S4–S6). We show that our energetic results obtained with the student M''_{16} potential are consistent with both Si-GAP-18 and Si-ACE-21.
- We improved our discussion of the relevance and expected impact of the work. In particular, we now highlight that our 100,000-atom models are realistic, as they correspond to length scales relevant to a-Si nanodevices. We also further describe the implications of our work for hydrogenated a-Si (a-Si:H), which could be a promising direction for future studies.

In the following, we quote all reviewers' comments in full, and our point-by-point response is interspersed in **blue**. Action taken is described in **red**.

Reviewer #1

This manuscript describes physically realistic modeling of amorphous silicon (a-Si) and paracrystallinity which may be present in the network as well as crystallization of the network. This type of modeling, validated by experimental observations, provides the framework for further investigations in the modifications of a-Si networks, such as through incorporation of hydrogen, dopant atoms, or group IV elements (C, Ge, Sn). The machine learning approach expands the capabilities of modeling to more closely describe larger scale systems, such as these amorphous networks. I recommend the manuscript be accepted for publication after the following points are addressed:

1) In the conclusion, application of this approach to other phenomena such as Staebler Wronski effect are mentioned. Hydrogenated amorphous silicon has been substantially studied, leading to other nomenclature such as “protocrystalline” and “edge” material. Descriptions of how these materials (and others) fit within the approach here should be added.

Response: This is a good suggestion, which we have implemented.

Action taken: We added the following text, together with relevant references (p. 12):

“Much research has been conducted on a-Si and a-Si:H, driven partly by photovoltaic applications. Following much interest in hydrogenated a-Si cells around 2000, there is now renewed focus on inexpensive a-Si:H cells for low-power applications, and as a component of tandem solar cells. So-called ‘protocrystalline’ photovoltaics are presumably hydrogenated variants of the paracrystalline phases that we study in the present paper, and they exhibit improved photo-stability relative to amorphous materials. Evidently, progress on the atomistic origins of the Staebler–Wronski effect requires large and realistic structural models. Indeed, a natural starting point could be to hydrogenate suitable models reported in the present paper, e.g., using a recently developed GAP ML potential for a-Si:H.”

2) Diamond like atomic environments are identified in the model with variations in percentages tracked with short and medium range order. With increased diamond like percentages, does the interconnectivity of the diamond like regions increase or do they remain relatively isolated? If interconnected, over what length scale?

Response: We thank the reviewer for this point. We have now analyzed the physical size and the interconnectivity of the paracrystalline clusters.

Action taken: We added a new figure to the main text (Fig. 3), which we introduce on p. 7. We also provide further information in the SI.

Journal Response Questions:

[What are the noteworthy results?] The noteworthy results consist of the development of a model spanning continuous random networks for amorphous silicon, paracrystallinity in silicon, and through to polycrystalline silicon. This includes energetic concerns relative to similarity of different networks to crystalline silicon, predicted medium range order compared to experiment, and structure factors. 100,000 atom models are developed so that the behavior of amorphous networks closer to that of reality ($> 10 \text{ nm} \times 10 \text{ nm} \times 10 \text{ nm}$) length scales can be evaluated.

[Will the work be of significance to the field and related fields? How does it compare to the established literature? If the work is not original, please provide relevant references.] This work is relevant to the field of amorphous silicon and amorphous materials in general. It provides physically realistic models that bridge the gaps from truly random networks through those that have some degree of crystallinity and shows that there is a continuum of behavior. It builds from established literature and previous work by these authors.

[Does the work support the conclusions and claims, or is additional evidence needed?] The results support the conclusions drawn. Theoretical models are validated by some experimental evidence. -[Are there any flaws in the data analysis, interpretation and conclusions? - Do these prohibit publication or require revision?] The previous comments involve more description of the literature in the amorphous silicon system and description of the isolation or interconnectivity of the diamond like regions remain to be addressed prior to publication.

[Is the methodology sound? Does the work meet the expected standards in your field?] The methodology builds from that established in the modeling of amorphous networks and expands it to larger size models. This is crucial to more realistically identifying the characteristics of disordered materials over the length scales where the materials are fabricated and utilized in applications.

[Is there enough detail provided in the methods for the work to be reproduced?] Yes, there is sufficient detail to reproduce the methodology.

Response: We thank the reviewer for this positive assessment.

Reviewer #2

This is an excellent paper of wide interest and importance as it addresses the fundamental structure of an important amorphous semiconductor. Silicon is also one of the most well-studied amorphous materials, yet ambiguity remains about the basic structure – is it a continuous random network (CRN) or does it have a higher degree of order? The authors, who are well regarded in the field of computational modeling and amorphous materials, use an innovative machine-learning approach to examine a very wide range of “quenched” structures and reveal an unexpected band of paracrystalline structures with a lower energy than the continuous random network. Their arguments are convincing that their work addresses a number of experimental results and adds substantial weight to the claim that the CRN is not the lowest energy state for amorphous silicon. I believe this work should be published in Nature Communications as it is a very fundamental and important new result.

Response: We thank the reviewer for their positive evaluation.

A couple of small questions to be considered:

a) Is figure 1 a plot of the enthalpy or the free energy?

Response: Figure 1 is a plot of the free energy per atom of each structure, relative to cubic diamond-type silicon.

Action taken: We revised the caption of Fig. 1 to state more clearly that we show “*the predicted excess free energy*” (p. 4).

b) While the modeling represents quenching from the melt, the comment about the deficit of structures between 5% and 14% crystal environments suggests that growth is occurring during the process. Are the authors able to comment on the presence or absence of a “critical nucleus size” for crystalline Si embedded in amorphous silicon?

Response: We thank the reviewer for this highly relevant comment. We believe that the slowest quench rate of 10^{10} K/s probes the limit between amorphization and crystallization. In our dataset, we see a large amount of very small subcritical clusters of < 10 atoms, then a jump to larger supercritical sizes of at least 40 atoms. We present the results of this analysis in a new main-text figure (Fig. 3), together with more details in the revised Supplementary Information.

Experimental work on homogeneous crystallization kinetics in a-Si suggests a critical nucleus size of around 40–60 atoms, equivalent to a spherical nucleus radius of around 0.6 nm.^{R2} Our dataset appears to corroborate the estimated experimental critical nucleus size, in as much as a direct comparison is possible, but the detailed mechanisms of crystallization could be the subject of its own study in the future.

Action taken: We added a discussion of the cluster sizes in relation to the critical nucleus, together with the new Fig. 3. We added Tables S2 and S3 to the Supplementary Information for further details.

c) I recognize that this is the first short publication of an exciting new result, but it would also be interesting to see some information on the physical size of the paracrystalline domains. This of course, could be covered in a future work - it's just a helpful (I hope!) suggestion.

Response: We thank the reviewer for this point. We have, accordingly, further analyzed the physical size and the connectivity of the paracrystalline clusters (in response to comments by both Reviewers #1 and #2).

Action taken: We added a new figure to the main text (Fig. 3), as mentioned above, and we provide further information in the Supplementary Information (Tables S2 and S3).

In this reviewer's opinion there is no need for subsequent review – could be published as is but the authors can consider my suggestions.

Reviewer #3

In this manuscript, the authors generated some configurations of a-Si containing different fractions of crystal- or diamond-like local environments, simply using different cooling rates and starting from different density, based on an MTP potential already published.

Then, they classified these configurations into three categories, i.e., fully disordered (CRN), partially crystallized (the so-called paracrystalline) and crystallized (polycrystalline), based on the thresholds they chose arbitrarily (as there is no any physical reason for their choice of threshold values in the manuscript).

First of all, I did not see that the current work presents any surprising results or new physical insight. It is well-known that the structure of any amorphous solids strongly depends on processing conditions. One even can find crystal-like local environments in undercooled liquids. In addition, when metastable amorphous solids crystallize into stable crystals, the change in structure (fraction of crystal-like environments) is definitely continuous. Therefore, one certainly can generate/observe amorphous configurations with different fractions of crystal-like local environments under different processing conditions either in real experiments or in simulations based on either ML or classical potentials. So, the main discovery presented in the current work is not a surprise.

Response: We thank the reviewer for their critical comments, which have shown us that we need to better justify the novelty of our work. We have therefore revised the manuscript to explain its relevance more clearly.

Our main discovery is not just the fact that a middle ground between true randomness and crystalline grains exists—noting that there previously existed a common belief of a configurational gap between these states^{R3,R4}—but also that this middle ground appears to provide a similar or even better match to calorimetric and structural data than the well-established CRN model alone. We think that this finding is relevant because the question of “CRN versus paracrystalline” has remained unresolved.

Action taken: We added references to recent publications arguing in favor of the CRN model^{R5} and conversely in favor of the paracrystalline model^{R6} to highlight that this question is still unresolved and not obvious (p. 3).

All the methodologies employed in the current work are already published in literature. Thus, this work did not present any new methodology.

Response: We agree with the reviewer that we do not develop any new methodologies in the current work—instead, we leverage the power of ML interatomic potentials to approach a long-standing open question.

So I can not support this work to publish on Nature Communications.

And below are some technical issues.

The authors argued that their dataset shown in Fig. 1 has high quality. But they just relaxed these configurations at 300 K for a very short time (10 ps) after fast quenching. These configurations are probably unstable or just some transient states. So why we need to care about the results based on this dataset?

Response: We thank the reviewer for this critical question. In response, we have performed further testing on the stability of our structures. We randomly chose 100 quenched structures from our dataset, and annealed them for 1 ns—that is, 100 times longer than in our initial protocol. We then compared the energetics and SOAP similarity values of the resulting structures to those of their 10-ps-annealed counterparts.

The results of this test show that annealing for 10 ps or 1 ns yields practically identical energetic and structural results. Based on these results, we argue that a 10 ps annealing simulation is sufficient for the purposes of the present study.

Action taken: We added a new Supplementary Figure to assess the influence of the annealing time (Fig. S2, p. S3).

According to Fig. 4, under the same cooling rate of $1e11$ K/s, using the two different potentials (GAP and MTP), the fraction of crystal-like atoms is different by 0.06%. One can understand this difference is due to artifact (the difference b/t the two ML potentials).

Response: We thank the reviewer for bringing this point to our attention—in fact, we found that we have made a typo in the labeling of the originally submitted Fig. 4. The structures have percentages of crystal-like atoms of 0.3% and 0.8%, respectively. The difference between the Si-GAP-18 and the M'_{16} generated structures is 0.5%, or 523 atoms.

Action taken: Figures 5 and S9 as well as the accompanying text (p. 10) have been modified to correct the typo. (The structures themselves have not changed.)

By the way, is there any difference in nucleation rate of silicon melts b/t the two ML potentials? For the GAP potential, nobody observed/reported crystallization from silicon melts. Interestingly, the authors observed crystallization from silicon melts with the MTP potential. Seems there is big difference in nucleation rate b/t the two ML potentials. If so, why should we believe the results based on the MTP potential. For silicon, an efficient ACE potential trained on the same dataset is also available, see npj Comput Mater 7, 97 (2021). Will the ACE potential give another different result?

Response: We thank the reviewer for suggesting the ACE potential from Ref. R1, to which we refer as “Si-ACE-21” in the following. This is an excellent way of placing our study on a wider footing, by including an additional relevant ML potential fitting methodology.

To verify the total and local atomic energy predictions which are central to our work (the claim of energetically favorable paracrystalline environments), we extended the analysis comparing M''_{16} and Si-GAP-18 to now include the Si-ACE-21 potential. All three potentials agree on the energy trends across the dataset (Fig. S4). They all suggest that the energetics for the paracrystalline structures are closer to the experimental ΔH than the those for the CRN structures, and that the distribution of atomic energies is smooth (Fig. S5).

Furthermore, we took a subset of 100 structures from our dataset, and relaxed each structure with all three potentials separately. Plotting the relative energies of the structures relaxed with M''_{16} and Si-ACE-21 against data for the same structures relaxed with Si-GAP-18 (Fig. S6), we see that Si-ACE-21 provides a closer match to the Si-GAP-18 relaxation, but that M''_{16} is still in reasonable agreement with both. The reviewer is correct, of course, in pointing out that different ML potentials will differ in the details of their predictions—however, the new ACE data allow us to quantify this difference more clearly, and to show that the overall *interpretation* is not affected by the particular choice of method. We think that this is a helpful message to add.

For further verification, we generated a new 100,000-atom model of a-Si using Si-ACE-21. We found 0.3% of diamond-like environments, in line with the $< 1\%$ of diamond-like environments obtained using Si-GAP-18 and M''_{16} .

In summary, the results obtained with M''_{16} are corroborated by the fact that two other models, the teacher Si-GAP-18 model and the Si-ACE-21 potential, lead to similar results. This substantiates our claim that the M''_{16} potential is valid.

Action taken: We added the new results to our comparative analysis between M''_{16} and Si-GAP-18, now also including Si-ACE-21, in the Supplementary Information (Sec. 1.2). We refer to this analysis in the Methods section (p. 13).

Actually, the difference $> 0.06\%$ in the fraction of crystal-like atoms can be observed in simulations with an identical potential and the same cooling history but quenching from different liquids at the same high temperature. Therefore, it is trivial to discuss two configurations with a difference by 0.06% in the fraction of crystal-like atoms.

Response: As discussed above (p. R7), we have now modified the text and figure to show the correct values.

When judging if an atom is crystal-like or not, they used PTM. This method is not very reliable as the results strongly depends on the cutoff of RMSD. The widely used method based on physically well-defined bond orientational order parameter, e.g., as used in Nat. Commun. 15, 368 (2024), should be more reliable.

Response: We thank the reviewer for suggesting the Steinhardt bond orientational order parameter (BOO), which is indeed a commonly used descriptor of

atomic neighborhoods. In response, we have now added an analysis using this BOO.

Specifically, we compare the prediction of different structural descriptors for the diamond-likeness of an environment by building a correlation heatmap (Fig. S8). Atomic SOAP similarity is the furthest from agreement with the others, while CNA and BOO similarity yield mediocre results. The poor performance by the atomic SOAP and BOO similarities arises from the absence of a consistent threshold of similarity to c-Si over the dataset, or even within a structure-type category, that provides good agreement—instead, the threshold must be chosen per structure, which we do not view as sufficiently robust for analysis. PTM, in contrast, yields the most consistent predictions. The classification of diamond-like environments and localized paracrystallinity could be the subject of its own study.

Action taken: We present the results of the comparative analysis of descriptors in Fig. S8, and reference it in the Methods section (p. 13).

Also, the choice for the threshold of fraction of crystal-like atoms used in separating paracrystalline and polycrystalline is pretty arbitrary.

Response: The threshold chosen for Polyhedral Template Matching is an important point—we thank the reviewer for highlighting this. To remove possible arbitrariness, we now systematically investigated how the PTM threshold influences the classification of structures into “paracrystalline” vs. “polycrystalline”, as presented in Sec. 1.3.1 of the Supplementary Information.

We show that at our chosen threshold of 15%, three quarters of the paracrystalline structures have fewer than 5% of diamond-like environments. Our analysis focuses on these structures with low paracrystallinity. Increasing the PTM threshold strongly, to 40%, only increases the percentage of paracrystalline structures by 3.2 points. Therefore, while the threshold is important, setting its value to 5% or to 40% would not substantially affect the results of our analysis.

Action taken: We present this additional analysis regarding the choice of the PTM threshold in Table S1 (p. S7) and Fig. S7 (p. S8). We point the reader to this analysis in the Methods section (p. 13).

The authors claimed their simulations with 100,000 atoms is realistic. But, in length scale, I feel billion-atom MD simulations, e.g., as conducted in Nat Commun 8, 10 (2017) by Shibuta, Y et. al., is more realistic.

Response: We thank the reviewer for this point, but we do believe that 100,000 atoms are sufficient in the specific case of a-Si studied herein. a-Si is used in a wide range of nanodevices of film thickness ranging from 10 nm for photodiodes^{R7} or light sensors^{R8} to layers of 100 nm for gravitational-wave detectors.^{R9,R10} Our models of 100,000 atoms correspond to simulation cells of about $12 \times 12 \times 12 \text{ nm}^3$, which we consider to be at device length scale.

Action taken: We modified the manuscript to emphasize the fact that the length scale of our models is “*relevant to a-Si-based devices*” (p. 10).

In conclusion, we thank all three reviewers again for their detailed and helpful feedback, which has allowed us to improve the manuscript further.

References

- R1. Lysogorskiy, Y. *et al.* Performant Implementation of the Atomic Cluster Expansion (PACE) and Application to Copper and Silicon. *npj Comput. Mater.* **7**, 97 (2021).
- R2. Spinella, C., Lombardo, S., Priolo, F. Crystal Grain Nucleation in Amorphous Silicon. *J. Appl. Phys.* **84**, 5383–5414 (1998).
- R3. Kail, F. *et al.* The configurational energy gap between amorphous and crystalline silicon. *Phys. Status Solidi RRL* **5**, 361–363 (2011).
- R4. Drabold, D. A. Silicon: The gulf between crystalline and amorphous. *Phys. Status Solidi RRL* **5**, 359–360 (2011).
- R5. Lévesque, C., Roorda, S., Schiettekatte, E., Mousseau, N. Internal Mechanical Dissipation Mechanisms in Amorphous Silicon. *Phys. Rev. Mater.* **6**, 123604 (2022).
- R6. Radić, D, Peterlechner, M., Posselt, M. , Bracht, H. Fluctuation Electron Microscopy on Amorphous Silicon and Amorphous Germanium. *Microsc. Microanal.* **29**, 477–489 (2023).
- R7. Street, R. A. Large Area Image Sensor Arrays. In *Technology and Applications of Amorphous Silicon*, 147-221 (Springer Berlin,, Heidelberg, Germany, 1999).
- R8. Street, R. A. Amorphous Silicon Device Technology. In *Hydrogenated Amorphous Silicon*, 363–403 (Cambridge University Press, Cambridge, United Kingdom, 1991).
- R9. Birney, R. *et al.* Amorphous Silicon with Extremely Low Absorption: Beating Thermal Noise in Gravitational Astronomy. *Phys. Rev. Lett.* **121**, 191101 (2018).
- R10. Adhikari, R. X. *et al.* A cryogenic silicon interferometer for gravitational-wave detection. *Class. Quantum Gravity* **37**, 165003 (2020).

Response to Reviewers' Comments for "Signatures of paracrystallinity in amorphous silicon"

Louise A. M. Rosset¹, David A. Drabold², and Volker L. Deringer¹

¹Department of Chemistry, University of Oxford, Oxford, UK

²Department of Physics and Astronomy, Ohio University, Athens, OH, USA

We thank all reviewers for their careful evaluation of our manuscript.

In the following, we quote all reviewers' comments in full, and our point-by-point response is interspersed in **blue**. Action taken is described in **red**.

Please note that the title of the manuscript has been changed to "Signatures of paracrystallinity in amorphous silicon *from machine-learning-driven molecular dynamics*" (additions emphasized) in the current revised submission.

Reviewer #1

The authors have satisfactorily addressed my previous comments. This manuscript is now ready for publication in my opinion.

The added information in the manuscript in the Discussion section addresses the comment regarding protocrystallinity and the Staebler Wronski effect.

The added information in the manuscript (Figure 3) and supplemental information (Section S2.4) addresses the previous comment regarding interconnectivity of diamond like regions.

Response: We thank the reviewer for their positive feedback.

Reviewer #3

Thanks the authors for clarifying the questions they are seeking to address.

From fully disordered to crystalline states, the structural variation, at least the time evolution of the fraction of local crystal-like environments, is definitely continuous, because it is impossible for an amorphous state to turn into crystalline state instantly, in configuration space. If this really needs a specific demonstration, I agree that the authors made it here. But I don't think that the results deserve a publication in Nature Communications. It might be a question long time ago, but now the answer is very obvious.

If the authors were demonstrating that there is a middle metastable state, i.e., so-called paracrystalline, between fully disordered and crystalline states, this would be a very interesting question. However, I do not think that the authors showed convincing evidence. To demonstrate it, one should show there is an additional basin in the middle of the free energy pathway from fully disordered to crystalline states, or show there is discontinuity in the evolution of one thermodynamic variable during the transitions from fully disordered to paracrystalline and from paracrystalline to crystalline.

Response: As argued by Reviewer #4 below, the paracrystallinity appears to originate from frozen subcritical nuclei (cf. Fig. 3) that persist as paracrystalline states at 300 K, but are unstable at higher temperatures (Supplementary Fig. 18). As there is substantial overlap between the paracrystalline configurational space and that of the CRN (Fig. 1), we view it as unlikely that a thermodynamic variable could show a discontinuity or an additional basin in the free-energy pathway between states.

The structural variation is continuous from fully disordered to crystalline states. I did not see any physical reason to pick a threshold value for a structural parameter to define an additional state, as the authors did in the manuscript. Following this logic, one could choose multiple thresholds to define multiple middle states in between fully disordered and crystalline states. Then more papers can be published.

Response: From Supplementary Fig. 7, we argue that the threshold that defines the intermediate states of para-/poly-crystallinity was chosen on a structural argument: beyond around 15% of diamond-like environments, we see clustering and a significant change in medium-range order—marking a difference in behavior between subcritical nuclei (paracrystalline) vs supercritical nuclei (polycrystalline).

If the authors are arguing that the amorphous model containing some fraction of local crystal-like environment shows better match to experimental data, I don't feel it is a big deal. Because one even can find small amount of local crystal-like environment in undercooled melts, it is natural to see some local crystal-like environments in glasses. It might be interesting to debate whether

amorphous silicon is fully disordered or so-called paracrystalline many decades ago, now it is well understood that glass structure depends on processing history. The current results might be suitable to publish on other journals, but not Nature Communications.

I doubted that configurations are probably unstable or just some transient states because they just relaxed 10 ps. Then the authors show there is no big difference even after relaxing for 1 ns. I agree that 1 ns is long for MD simulations especially those based on ML potentials. But 1 ns is far shorter than physical time for real materials. It may be hard to use straightforward MD simulations to give a convincing answer.

Response: While we agree that 1 ns is far shorter than the timescale of many physical processes in materials, this is due to intrinsic limitations of atomistic simulations. The usefulness of nanosecond-scale MD simulations for modelling amorphous silicon has been widely documented in the literature (see, e.g., Refs. R1–R4).

The authors argue that BOO is not as good as PTM in identifying diamond-like environment from amorphous silicon. But the authors did not provide sufficient detail of the BOO they used. And I did see there is a big gap in the distribution of a single BOO between amorphous silicon and diamond crystals in the reference I mentioned in my first comments.

Action taken: We added further details of the BOO implementation on page S11 of the Supplementary Information.

Reviewer #4

Per the Editor's request, I will focus exclusively on the review comments provided by the third reviewer and will refrain from making recommendations on the manuscript's suitability for publication in Nature Communications. Regarding the third reviewer's questions, I will specifically address the following: Is thermodynamic quantification of the paracrystalline state necessary beyond the PTM method?

The reviewer raises an important point: demonstrating that the paracrystalline state is a thermodynamically metastable phase (e.g., through free energy analysis) would indeed constitute groundbreaking work. That said, this manuscript approaches the issue from a structural perspective. The authors present a realistic modeling framework to interpret the structure of amorphous silicon (a-Si), introducing the concept of paracrystallinity or medium-range crystalline order as an extended configuration of the amorphous phase (as opposed to the widely accepted CRN structure).

The observed paracrystallinity likely originates from a nucleation process with a vanishingly small nucleation barrier in the supercooled liquid state. Due to kinetic constraints, subcritical nuclei (comprising tens of atoms) are frozen in local energy minima, persisting as a metastable "paracrystalline" state at 300 K. This state is unstable at elevated temperatures (e.g., near the glass transition temperature). It is unlikely that a thermodynamic variable exists to distinguish the paracrystalline state as an intermediate phase with a discrete energy minimum separating the crystalline and continuous random network (CRN)-type amorphous phases.

From a thermodynamic perspective, the paracrystalline state extends the configurational space of CRN-type a-Si. Transitions to polycrystalline Si require thermal activation. This paracrystalline state may persist for extended timescales at 300 K (e.g., 10 ps, 10 ns, or longer), but it remains metastable under these conditions and transitions to instability at higher temperatures (i.e., slower cooling rates).

In summary: The paracrystalline structure of a-Si results from a kinetically constrained nucleation process. It is metastable at 300 K but becomes unstable at elevated temperatures. The existence of a thermodynamic variable that could definitively classify the paracrystalline state as an intermediate phase is highly unlikely. Within this context, the authors' efforts to interpret the structure seem sufficient.

Response: We thank the reviewer for their points, which summarize our findings from Fig. 3, Supplementary Note 2.D, and Supplementary Fig. 18.

Lastly, I observed that the MTP potential shows noticeable energy discrepancies compared to the other potentials used (Figure S5).

Response: The M''_{16} potential is a ‘student model’ of the Si-GAP-18 potential. As described in Ref. R5, the Si-GAP-18 ‘teacher model’ was used to run MD simulations and label structures from those with Si-GAP-18 energies and forces. The M''_{16} potential was then fit to the resulting dataset. M''_{16} is a much faster model than its ‘teacher’, but it has slightly reduced accuracy because it is based on synthetic labels, rather than directly on DFT data^{R5}. This implies that there exists a trade-off between speed and accuracy when using the M''_{16} potential.

In the present study, computational speed is key to access the 10^{10} K/s quench rate and to sample the wide-ranging configurational space from disorder to order (Supplementary Fig. 3). Previous work in Ref. R5 provides support for the suitability of M''_{16} -driven MD for modeling a-Si (in particular, by analyzing the computed structure factor, shown in Figs. S3 and S4 of the Supplementary Material of Ref. R5).

When evaluated on the same structures, the M''_{16} potential’s predictions of per-atom energy distributions compare well with predictions from both Si-GAP-18 and Si-ACE-21 (Supplementary Figs. 4 and 5). In particular for Supplementary Fig. 5, the shape of the distribution predicted by M''_{16} aligns with that predicted by Si-GAP-18. We do note on p. S8 that the M''_{16} model “*over-stabilizes the highly distorted CRN structures*” compared to Si-GAP-18 in Supplementary Fig. 6, which tests the description of fully relaxed local minima. Even in this case, the overall stability trends seem consistent across all three models.

In conclusion, we thank all reviewers again for their helpful comments and suggestions.

References

- R1. Sastry, S. & Angell, C. A. Liquid–liquid phase transition in supercooled silicon. *Nat. Mater.* **2**, 739–743 (2003).
- R2. Deringer, V. L. *et al.* Realistic Atomistic Structure of Amorphous Silicon from Machine-Learning-Driven Molecular Dynamics. *J. Phys. Chem. Lett.* **9**, 2879–2885 (2018).
- R3. Deringer, V. L. *et al.* Origins of structural and electronic transitions in disordered silicon. *Nature* **589**, 59–64 (2021).
- R4. Fan, Z. & Tanaka, H. Microscopic mechanisms of pressure-induced amorphous-amorphous transitions and crystallisation in silicon. *Nat. Commun.* **15**, 368 (2024).
- R5. Morrow, J. D. & Deringer, V. L. Indirect learning and physically guided validation of interatomic potential models. *J. Chem. Phys.* **157**, 104105 (2022).